



**Reviewing Global Estimates of Surface Reactive Nitrogen Concentration and**
**Deposition Using Satellite Observation**
Lei Liu [a, *], Xiuying Zhang [b], Wen Xu [c], Xuejun Liu [c], Xuehe Lu [b], Jing Wei [d, e], Yi Li [f],
Yuyu Yang [a], Zhen Wang [b], Anthony Y. H. Wong [g]
[a] College of Earth and Environmental Sciences, Lanzhou University, Lanzhou 730000,
China
[b] International Institute for Earth System Science, Nanjing University, Nanjing,
210023, China
[c] College of Resources and Environmental Sciences, National Academy of
Agriculture Green Development, China Agricultural University, Beijing, 100193,
China
[d] State Key Laboratory of Remote Sensing Science, College of Global Change and
Earth System Science, Beijing Normal University, Beijing, China
[e] Department of Atmospheric and Oceanic Science, Earth System Science
Interdisciplinary Center, University of Maryland, College Park, MD, USA
[f] Chief Technology Officer SailBri Cooper Inc., Beaverton OR, 97008, USA
[g] Department of Earth and Environment, Boston University, Boston, MA 02215, USA
* Correspondence to Lei Liu (liuleigeo@lzu.edu.cn).
**Abstract**
Since the industrial revolution, human activities have dramatically changed the
nitrogen (N) cycle in natural systems. Anthropogenic emissions of reactive nitrogen
($N_r$) can return to the earth's surface through atmospheric $N_r$ deposition. Increased $N_r$
deposition may improve ecosystem productivity. However, excessive $N_r$ deposition
can cause a series of negative effects on ecosystem health, biodiversity, soil, and
water. Thus, accurate estimations of $N_r$ deposition are necessary for evaluating its





environmental impacts. The United States, Canada and Europe have successively
launched a number of satellites with sensors that allow retrieval of atmospheric $NO_2$
and $NH_3$ column density, and therefore estimation of surface $N_r$ concentration and
deposition at an unprecedented spatiotemporal scale. Atmosphere $NH_3$ column can be
retrieved from atmospheric infra-red emission measured by IASI, AIRS, CrIS or TES,
while atmospheric $NO_2$ column can be retrieved from reflected solar radiation
measured by GOME, GOME-2, SCIAMACHY, OMI, TEMPO, Sentinel and GEMS.
In recent years, scientists attempted to estimate surface $N_r$ concentration and
deposition using satellite retrieval of atmospheric $NO_2$ and $NH_3$ columns. In this study,
we give a thorough review on recent advances of estimating surface $N_r$ concentration
and deposition using the satellite retrievals of $NO_2$ and $NH_3$, present a framework of
using satellite data to estimate surface $N_r$ concentration and deposition based on
recent works, and summarize the existing challenges for estimating surface $N_r$
concentration and deposition using the satellite-based methods. We believe that
exploiting satellite data to estimate $N_r$ deposition has a broad and promising prospect.
**Keywords**
Nitrogen deposition; Satellite retrieval; Surface concentration; Oxidized and reduced
$N_r$
**1. Introduction**
Nitrogen (N) exists in three forms in the environment including reactive nitrogen ($N_r$),
organic nitrogen (ON) and nitrogen gas ($N_2$) (Canfield et al., 2010). $N_2$ is the main
component of air, accounting for 78% of the total volume of air, but it cannot be
directly used by most plants. $N_r$ (such as $NO_3^-$ and $NH_4^+$) is the main form of N that
can be directly used by most plants, but the content of $N_r$ in nature is much lower
compared with ON and $N_2$ (Vitousek et al., 1997;Nicolas and Galloway, 2008). The



supply of $N_r$ is essential for all life forms and contributes to the increase in
agricultural production, thus providing sufficient food for the growing global
population (Galloway et al., 2008;David et al., 2013;Galloway et al., 2004b;Erisman
et al., 2008). Before the industrial revolution, $N_r$ mainly came from natural sources
such as biological N fixation, lightning and volcanic eruption (Galloway et al., 2004a).
Since the industrial revolution, human activities (e.g. agricultural development,
combustion of mineral energy) have greatly perturbed the N cycle in natural systems
(Canfield et al., 2010;Kim et al., 2014;Lamarque et al., 2005).
$N_r$ ($NO_x$ and $NH_3$) emitted to the atmosphere will return to the earth surface through
atmospheric deposition (Liu et al., 2011). Atmospheric $N_r$ deposition refers to the
process in which $N_r$ are removed from the atmosphere, including wet (rain and snow)
and dry (gravitational settling, atmospheric turbulence, etc.) deposition (Xu et al.,
2015;Zhang et al., 2012;Pan et al., 2012). The input of $N_r$ over terrestrial natural
ecosystems primarily comes from the $N_r$ deposition (Shen et al., 2013;Sutton et al.,
2001;Larssen et al., 2011). In the short term, atmospheric $N_r$ deposition can increase
the $N_r$ input to ecosystems, which promotes plant growth and enhances ecosystem
productivity (Erisman et al., 2008;Sutton et al., 2013). However, excessive
atmospheric $N_r$ deposition also causes a series of environmental problems (Liu et al.,
2017d). Due to the low efficiency of agricultural N application, plenty of $N_r$ is lost
through runoff, leaching and volatilization, causing serious environmental pollution.
Excessive $N_r$ deposition may aggravate the plant's susceptibility to drought or frost,
reduce the resistance of plant to pathogens or pests, and further affect the physiology
and biomass distribution of vegetation (ratio of roots, stems and leaves) (Stevens et al.,
2004;Nadelhoffer et al., 1999;Bobbink et al., 2010;Janssens et al., 2010). Excessive
$N_r$ leads to eutrophication and related algal blooms over aquatic ecosystems, reducing





water biodiversity (Paerl et al., 2014), while excessive $N_r$ in drinking water also poses
a threat to human health (Zhao et al., 2013). Therefore, monitoring and estimation of
surface $N_r$ concentration and deposition on the global scale are of great importance
and urgency.
The methods of estimating atmospheric $N_r$ deposition can be divided into three
categories: ground-based monitoring, atmospheric chemical transport modeling
(ACTM) and satellite-based estimation. Ground-based monitoring is considered to be
the most accurate quantitative method, which can effectively reflect the $N_r$ deposition
in local areas. ACTM can simulate the processes of $N_r$ chemical reaction, transport,
and deposition, as well as the vertical distribution of $N_r$. Satellite-based estimation
establishes empirical, physical or semi-empirical models by connecting the
ground-based $N_r$ concentrations and deposition with satellite-derived $N_r$ concentration.
This study focuses on reviewing the recent development of satellite-based methods to
estimate $N_r$ deposition. We firstly give a brief introduction to the progress of
ground-based monitoring, ACTM-based methods, and then present a detailed
framework of using satellite observation to estimate dry and wet $N_r$ deposition
(including both oxidized and reduced $N_r$). Next, we review the recent advances of the
satellite-based methods of estimating $N_r$ deposition. Finally, we discuss the remaining
challenges for estimating surface $N_r$ concentration and deposition using satellite
observation.
**2.1 Methods for Estimating Surface $N_r$ Concentration and Deposition**
**2.1.1 Ground-based Monitoring**
Ground-based monitoring of $N_r$ deposition can be divided into two parts: wet and dry
$N_r$ deposition monitoring. Since the 1970s, there have been large-scale monitoring
networks focusing on the wet $N_r$ deposition. The main large-scale regional monitoring



networks include Canadian Air and Precipitation Monitoring Network (CAPMoN),
Acid Deposition Monitoring Network in East Asia (EANET), European Monitoring
and Evaluation Program (EMEP), United States National Atmospheric Deposition
Program (NADP), World Meteorological Organization Global Atmosphere Watch
Precipitation Chemistry Program, and Nationwide Nitrogen Deposition Monitoring
Network in China (NNDMN) (Tan et al., 2018;Vet et al., 2014). The detailed
scientific objectives of the wet $N_r$ deposition observation networks vary, but most of
the observation networks mainly concentrate on the spatiotemporal variation of wet
deposition of ions including $N_r$ compounds, the long-term trends of ions in
precipitation, and the evaluation of ACTMs.
Compared with wet $N_r$ deposition monitoring, dry $N_r$ deposition monitoring started
late, due to the limitation of monitoring technology since it is more difficult to be
quantified (affected greatly by surface roughness, air humidity, climate and other
environmental factors) (Liu et al., 2017c). Dry $N_r$ deposition observation networks
include US ammonia monitoring network (AMoN), CAPMoN, EANET and EMEP.
The monitoring methods of dry $N_r$ deposition are mainly divided into direct
monitoring (such as dynamic chambers) and indirect monitoring (such as inferential
methods). The inferential model is widely applied in ground-based monitoring
networks (such as EANET and NNDMN), mainly because this method is more
practical and simpler. In inferential models, dry deposition is divided into two parts:
surface $N_r$ concentrations and the deposition velocity ($V_d$) of $N_r$ (Nowlan et al., 2014).
$V_d$ can be estimated by meteorology, land use types of underlying surface as well as
the characteristics of each $N_r$ component itself using resistance models (Nemitz et al.,
2001). Thus, dry $N_r$ deposition monitoring networks only need to focus on the
quantification of surface concentration of individual $N_r$ components. The $N_r$





components in the atmosphere are very complex, including $N_2O_5$, HONO, $NH_3$, $NO_2$,
$HNO_3$ and particulate $NH_4^+$ and $NO_3^-$. Most monitoring networks include the major
$N_r$ species such as gaseous $NH_3$, $NO_2$, $HNO_3$ and the particles of $NH_4^+$ and $NO_3^-$.
Effort of ground-based $N_r$ deposition monitoring mostly concentrates on wet $N_r$
deposition, while observations of dry $N_r$ deposition are relatively scarce especially for
surface $HNO_3$ and $NH_4^+$ and $NO_3^-$. Second, most observation networks focus on a few
years or a certain period of time, leading to the lack of long-term continuously
monitoring on both wet and dry $N_r$ deposition. More importantly, the global $N_r$
deposition monitoring network has not been established, and the sampling standards
in different regions are not unified. These outline the potential room for improvement
of ground-based $N_r$ deposition monitoring.
**2.1.2 ACTM Simulation**
An ACTM can simulate $N_r$ deposition at regional or global scales through explicitly
representing the physical and chemical processes of atmospheric $N_r$ components
(Zhao et al., 2017;Zhang et al., 2012). Wet $N_r$ deposition flux is parameterized as
in-cloud, under-cloud and precipitation scavenging (Amos et al., 2012;Levine and
Schwartz, 1982;Liu et al., 2001;Mari et al., 2000), while dry deposition flux can be
obtained as the product of surface $N_r$ concentration and $V_d$, which is typically
parameterized as a network of resistances (Wesely and Hicks, 1977). Based on the
integrated results of 11 models of HTAP (hemispheric transport of air pollution), Tian
et al. found that about 76%-83% of the ACTM's simulation results were ±50% of the
monitoring values, and the modeling results underestimated the wet deposition of
$NH_4^+$ and $NO_3^-$ over Europe and East Asia, and overestimated the wet deposition of
$NO_3^-$ over the eastern US. Though regional ACTMs can be configured at very high
horizontal resolution (e.g., $1 \times 1$ $km^2$) (Kuik et al., 2016), the horizontal resolution of





global ACTMs are relatively coarse $(1°\times1°-5°\times4°)$ (Williams et al., 2017), which
cannot indicate the local pattern of $N_r$ deposition. On the other hand, the $N_r$ emission
inventory used to drive an ACTM is highly uncertain, with the uncertainty of the $NO_x$
emission at about $\pm30\text{-}40\%$, and that of $NH_3$ emission at about $\pm30\text{-}80\%$ (Zhang et al.,
2009;Cao et al., 2011).
**2.1.3 Satellite-based Estimation of Surface $N_r$ Concentration and Deposition**
Satellite observation has wide spatial coverages and high resolution, and is
spatiotemporally continuous. Atmospheric $NO_2$ and $NH_3$ columns can be derived
from satellite measurements with relatively high accuracy (Van Damme et al.,
2014a;Boersma et al., 2011), providing a new perspective about atmospheric $N_r$
abundance.
Satellite instruments that can monitor $NO_2$ in the atmosphere include GOME (Global
Ozone Monitoring Experience), SCIAMACHY (SCanning Imaging Absorption
SpectroMeter for Atmospheric ChartographY), OMI (Ozone Monitoring Instrument),
GOME-2 (Global Ozone Monitoring Experience-2). Some scholars applied satellite
$NO_2$ columns to estimate the surface $NO_2$ concentration, and then dry $NO_2$ deposition
by combining the surface $NO_2$ concentration and modeled $V_d$. Cheng et al. (Cheng et
al., 2013) established a statistical model to estimate the surface $NO_2$ concentration
based on the SCIAMACHY $NO_2$ columns, and then estimated the dry deposition of
$NO_2$ over eastern China. This method by Cheng et al. (Cheng et al., 2013) using the
simple linear model did not consider the vertical profiles of $NO_2$. Lu et al. (Lu et al.,
2013) established a multivariate linear regression model based on the SCIAMACHY
and GOME $NO_2$ columns, meteorological data and ground-based monitoring $N_r$
deposition, and then estimated the global total $N_r$ deposition. Lu et al. (Lu et al., 2013)
could not distinguish the contribution of dry and wet $N_r$ deposition using the





multivariate linear regression model. Jia et al. (Jia et al., 2016) established a simple
linear regression model based on OMI tropospheric $NO_2$ column and ground-based
surface $N_r$ concentration, and then estimated the total amounts of dry $N_r$ deposition.
Jia et al. (Jia et al., 2016) used OMI tropospheric $NO_2$ column to estimate the dry
deposition of reduced $N_r$ deposition ($NH_3$ and $NH_4^+$), which could also bring great
errors since the OMI $NO_2$ column could not indicate the $NH_3$ emission. These studies
highlight the problem of using only $NO_2$ columns to derive total $N_r$ deposition, that
$NO_2$ columns give us highly limited information about the abundance of reduced $N_r$
($NH_3$ and $NH_4^+$).
Lamsal et al. (Lamsal et al., 2008) first used the relationship between the $NO_2$ column
and surface $NO_2$ concentration at the bottom layer simulated by an ACTM to convert
OMI $NO_2$ column to surface $NO_2$ concentration. A series of works (Lamsal et al.,
2013;Nowlan et al., 2014;Kharol et al., 2018) have effectively estimated regional and
global surface $NO_2$ concentration using satellite $NO_2$ column combining with
ACTM-derived relationship between the $NO_2$ column and surface $NO_2$ concentration
simulated. It is worth mentioning that Nowlan et al. (Nowlan et al., 2014) applied
OMI $NO_2$ column to obtain the global dry $NO_2$ deposition during 2005-2007 for the
first time. However, using satellite $NO_2$ column and ACTM-derived relationship
between the $NO_2$ column and surface $NO_2$ concentration may lead to an
underestimation of surface $NO_2$ concentration. Kharol et al. (Kharol et al., 2015)
found that the satellite-derived surface $NO_2$ concentration using the above method is
only half of the observed values. To resolve such potential underestimation, Larkin et
al. (Larkin et al., 2017) established a statistical relationship between the
satellite-derived and ground measured surface $NO_2$ concentration, and then calibrated
the satellite-derived surface $NO_2$ concentration using the established relationship.



Some researchers also estimated other $N_r$ components (such as particulate $NO_3^-$)
based on satellite $NO_2$ column. Based on the linear model between $NO_2$, $NO_3^-$, $HNO_3$
obtained by ground-based measurements, Jia et al. (Jia et al., 2016) calculated the
surface $NO_3^-$ and $HNO_3$ concentration using satellite-derived surface $NO_2$
concentration and their relationship. Geddes et al. (Geddes and Martin, 2017)
reconstructed the $NO_x$ emission data by using the satellite $NO_2$ column, and then
estimated the global $NO_x$ deposition by an ACTM, but the spatial resolution of global
$NO_x$ deposition remains low ($2°\times2.5°$), failing to exploit the higher resolution of
satellite observation.
Comparing with $NO_2$, the development of satellite $NH_3$ monitoring is relatively late.
Atmospheric $NH_3$ was first detected by the TES in Beijing and Los Angeles (Beer et
al., 2008). The IASI sensor also detected atmospheric $NH_3$ from a biomass burning
event in Greece (Coheur et al., 2009). Subsequently, many scholars began to develop
more reliable satellite $NH_3$ column retrievals (Whitburn et al., 2016;Van Damme et al.,
2014a), validate the satellite-retrieved $NH_3$ column with the ground-based observation
(Van Damme et al., 2014a;Dammers et al., 2016;Li et al., 2017), and compare the
satellite $NH_3$ column with the aircraft measured $NH_3$ column (Van Damme et al.,
2014b;Whitburn et al., 2016). In recent years, some scholars have carried out the
works of estimating surface $NH_3$ concentration based on satellite $NH_3$ column. Liu et
al. (Liu et al., 2017b) obtained the satellite-derived surface $NH_3$ concentration in
China based on the IASI $NH_3$ column coupled with an ACTM, and deepened the
understanding of the spatial pattern of surface $NH_3$ concentration in China. Similarly,
Graaf et al. (Van der Graaf et al., 2018) carried out the relevant work in Europe based
on the IASI $NH_3$ column coupled with an ACTM, and estimated the dry $NH_3$
deposition in West Europe. Jia et al. (Jia et al., 2016) first constructed the linear model


between surface $NO_2$ and $NH_4^+$ concentration based on ground monitoring data, and
then calculated the $NH_4^+$ concentration using satellite-derived surface $NO_2$
concentration and their relationship. However, as the emission sources of $NO_x$
(mainly from transportation and energy sectors) and $NH_3$ (mainly from agricultural
sector) are different (Hoesly et al., 2018), the linear model between surface $NO_2$ and
$NH_4^+$ concentration may lead to large uncertainties in estimating the global $NH_4^+$
concentration. There is still no report about the satellite-derived dry and wet reduced
$N_r$ deposition using satellite $NH_3$ column at a global scale. As reduced $N_r$ plays an
important role in total $N_r$ deposition, satellite $NH_3$ should be better utilized to help
estimate reduced $N_r$ deposition.
**2.1.4 Problems in Estimating Global $N_r$ Deposition**
The spatial coverage of ground monitoring sites focusing on $N_r$ deposition is still not
adequate, and the monitoring standards and specifications in different regions of the
world are not consistent, presenting a barrier to integrating different regional
monitoring data. Large uncertainties exist in $N_r$ emission inventory used to drive the
ACTMs, and the spatial resolution of the modeled $N_r$ deposition by ACTMs is coarse.
Using satellite monitoring data to estimate surface $N_r$ concentration and deposition is
still in its infancy, especially for reduced $N_r$.
Some scholars tried to use satellite $NO_2$ and $NH_3$ column to estimate the surface $N_r$
concentration and dry $N_r$ deposition. However, there are relatively few studies on
estimating wet $N_r$ deposition. In addition, the development of satellite monitoring for
$NH_3$ in the atmosphere is relatively late (compared with $NO_2$). At present, IASI $NH_3$
data have been widely used, while the effective measurements of TES are less than
IASI; CrIS and AIRS $NH_3$ column products are still under development. There are
three main concerns in high-resolution estimation of surface $N_r$ concentration and





deposition based on satellite $N_r$ observation. (1) How to effectively couple the satellite
high-resolution $NO_2$ and $NH_3$ column data with the vertical profiles simulated by an
ACTM, and then estimates the surface $N_r$ concentrations? This step is the key to
simulate the dry $N_r$ deposition. (2) How to construct a model for estimating dry $N_r$
deposition including all major $N_r$ species based on satellite $NO_2$ and $NH_3$ column, and
then estimates the dry $N_r$ deposition at a high spatial resolution? (3) How to combine
the high-resolution satellite $NO_2$ and $NH_3$ column data and ground-based monitoring
data to construct wet $N_r$ deposition models, and then estimate the wet $N_r$ deposition at
a high spatial resolution?
**3. Framework of Estimating Surface $N_r$ Concentration and Deposition Using**
**Satellite Observation**
We give a framework of using satellite observation to estimate surface $N_r$
concentration and deposition as shown in **Fig. 1** based on recent advances.

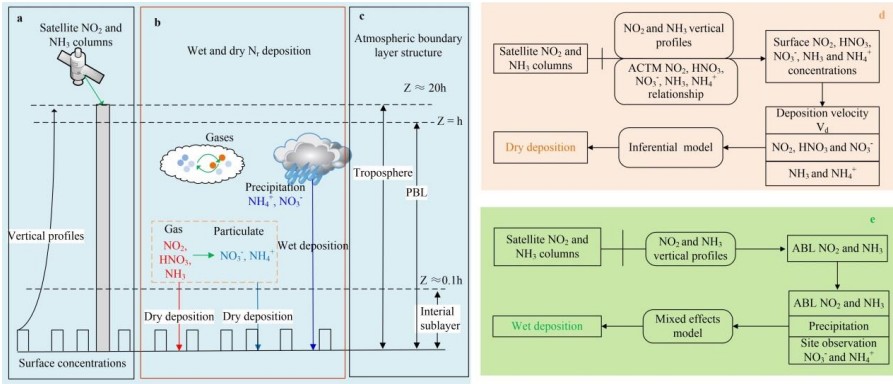


**Fig. 1 Schematic diagram of dry and wet $N_r$ deposition.** (a) indicates satellite observed $NO_2$
and $NH_3$ column, and the vertical profiles by an ACTM; (b) shows dry and wet $N_r$ deposition
including the major $N_r$ species (gaseous $NO_2$, $HNO_3$, $NH_3$, particulate $NO_3^-$ and $NH_4^+$, as well as
wet $NO_3^-$ and $NH_4^+$ in precipitation); (c) illustrates atmospheric vertical structures including the
troposphere (satellite observation), atmospheric boundary layer (ABL), interfacial sub-layer; (d)
and (e) represent procedures of calculating the dry and wet $N_r$ deposition.

**3.1.1 Conversion of Satellite $NO_2$ and $NH_3$ Column to Surface $N_r$ Concentration**
An ACTM can simulate the vertical profiles of $NO_2$ and $NH_3$ with multiple layers



from the surface to the troposphere. For example, the GEOS-Chem ACTM includes
47 vertical layers from the earth surface to the top of the stratosphere. Most previous
studies estimated the ratio of surface $N_r$ concentration (at the first layer) to total
columns by an ACTM, and then multiply the ratio by satellite columns to estimate
satellite-derived surface concentration (Geddes et al., 2016;Graaf et al., 2018;Nowlan
et al., 2014).
Another approach tries to fit general vertical profiles of $NO_2$ and $NH_3$ (Zhang et al.,
2017;Liu et al., 2017b;Liu et al., 2017c), and then estimate the ratio of $N_r$
concentration at any height to total $N_r$ columns, and finally multiply the ratio by
satellite $NO_2$ and $NH_3$ columns. This approach has an advantage compared with the
previous one for that $NO_2$ and $NH_3$ concentration at all altitude included in ACTM
simulations can be estimated.
Taking the estimation of surface $NO_2$ concentration using the latter approach as an
example, the methods and steps are introduced in the following.
Step 1: Calculate the monthly mean $NO_2$ concentrations at all layers simulated by an
ACTM.
Step 2: Construct the vertical profile function of $NO_2$. Multiple Gaussian functions are
used to fit the vertical distribution of $NO_2$ based on the monthly $NO_2$ concentrations at
all layers calculated in Step 1, in which the independent variable is the height
(altitude), and the dependent variable is $NO_2$ concentration at a certain height.
The basic form of single Gaussian function is (Zhang et al., 2017;Liu et al., 2017b;Liu
et al., 2017c;Whitburn et al., 2016):
$$\rho = \rho_{max} e^{-(\frac{Z-Z_0}{\sigma})^2} \quad (1)$$
where Z is the height of a layer in the ACTM; $\rho_{max}$, $Z_o$ and $\sigma$ are the maximum $NO_2$
concentration, the corresponding height with the maximum $NO_2$ concentration and the





299 thickness of $NO_2$ concentration layer (one standard error of Gaussian function).

300 There are two basic forms of profile shapes of $NO_2$: (1) $NO_2$ concentration reaches the

301 maximum concentration when reaching a certain height ($Z_o \neq 0$). As the height

302 increases, the $NO_2$ concentration begins to decline; (2) $NO_2$ concentration is basically

303 concentrated on the earth surface ($Z_o = 0$). These two cases are the ideal state of the

304 vertical distribution of $NO_2$ concentration. In reality, single Gaussian fitting may not

305 capture the vertical distribution of $NO_2$ well. To improve the accuracy of fitting, the

306 sum of multiple Gaussian functions can be used:

307 $\rho(Z) = \sum_{i=1}^{n} \rho_{max,i} e^{-(\frac{Z-Z_{0,i}}{\sigma_i})^2}$ (2)

308 Step 3: Calculate the ratio of $NO_2$ concentration at the height of $h_G$ to total columns

309 ($\int_0^{h_{trop}} \rho(Z)\,dx$), and then multiply the ratio by satellite column ($S_{trop}$). The

310 satellite-derived $N_r$ concentration at the height of $h_G$ can be calculated as:

311 $S_{G\_NO2} = S_{trop} \times \frac{\rho(h_G)}{\int_0^{h_{trop}} \rho(Z)\,dx}$ (3)

312 Step 4: Convert the instantaneous satellite-derived surface $NO_2$ concentration ($S_{G\_NO2}$)

313 to daily average ($S_{G\_NO2}*$) using the ratio of average surface $NO_2$ concentration

314 ($G_{ACTM}^{1-24}$) to that at satellite overpass time ($G_{ACTM}^{overpass}$) by an ACTM:

315 $S_{G\_NO2}* = \frac{G_{ACTM}^{1-24}}{G_{ACTM}^{overpass}} \times S_{G\_NO2}$ (4)

316 The method for estimating the surface $NH_3$ concentration ($S_{G\_NH3}*$) is similar to that

317 for estimating the surface $NO_2$ concentration.

318 **3.1.2 Estimating Surface Concentration of Other $N_r$ Species**

319 At present, only $NO_2$ and $NH_3$ column can be retrieved reliably, and there are no

320 reliable satellite retrievals of $HNO_3$, $NH_4^+$ and $NO_3^-$. For example, the IASI $HNO_3$

321 product is still in the stage of data development and verification (Ronsmans et al.,



2016). Previous studies firstly derive the relationship between $N_r$ species by an
ACTM or by ground-based measurements, and then use the relationship to convert
satellite-derived surface $NO_2$ and $NH_3$ concentration ($S_{G\_NH3}$ *) to $HNO_3$, $NH_4^+$ and
$NO_3^-$ concentrations:
$$
\begin{cases}
G_{S\_NO3} = S_{G\_NO2} * \times \frac{G_{ACTM\_NO3}}{G_{ACTM\_NO2}} \\
G_{S\_HNO3} = S_{G\_NO2} * \times \frac{G_{ACTM\_HNO3}}{G_{ACTM\_NO2}} \quad (5) \\
G_{S\_NH4} = S_{G\_NH3} * \times \frac{G_{ACTM\_NH4}}{G_{ACTM\_NH3}}
\end{cases}
$$

$\frac{G_{ACTM\_NO3}}{G_{ACTM\_NO2}}, \frac{G_{ACTM\_HNO3}}{G_{ACTM\_NO2}}, \frac{G_{ACTM\_NH4}}{G_{ACTM\_NH3}}$ is the estimated ratio of between $NO_2$ and $NO_3^-$,
$NO_2$ and $HNO_3$, $NH_3$ and $NH_4^+$.
**3.1.3 Dry Deposition of $N_r$**
The resistance of dry $N_r$ deposition mainly comes from three aspects: aerodynamic
resistance ($R_a$), quasi laminar sub-layer resistance ($R_b$) and canopy resistance ($R_c$).
The $V_d$ can be expressed as
$V_d = \frac{1}{R_a + R_b + R_c} + v_g \quad (6)$
$V_g$ is gravitational settling velocity. For gases, the $V_g$ is negligible ($V_g=0$).
Dry $NO_2$, $NO_3^-$, $HNO_3$, and $NH_4^+$ deposition can be calculated by:
$F = G_S \times V_d \quad (7)$
Unlike above species, $NH_3$ is bi-directional, presenting both upward and downward
fluxes. There is a so-called "canopy compensation point" ($C_o$) controlling dry $NH_3$
deposition. Dry $NH_3$ deposition can be calculated by:
$F = (G_{S\_NH3} - C_o) \times V_d \quad (8)$
The calculation of $C_o$ is very complex including the leaf stomatal and soil emission
potentials related to the meteorological factors, the plant growth stage and the canopy
type. The satellite-based methods usually neglected this complex process and set $C_o$





as zero (Graaf et al., 2018;Kharol et al., 2018) or set fixed values in each land use
type based on ground-based measurements (Jia et al., 2016).
**3.1.4 Wet Deposition of $N_r$**
The satellite-based estimation of wet $N_r$ deposition can be simplified as the product of
the concentration of $N_r$ (C), precipitation (P) and scavenging coefficient (w) (Pan et
al., 2012). Satellite $NO_2$ and $NH_3$ can be used to indicate the oxidized $N_r$ and reduced
$N_r$; precipitation (P) can be obtained from ground monitoring data or reanalysis data
(such as NCEP). However, the scavenging coefficient (w) is usually highly uncertain.
To improve the accuracy of estimation, a mixed-effects model (Liu et al.,
2017a;Zhang et al., 2018) is proposed to build the relationship between satellite $NO_2$
and $NH_3$, precipitation and ground monitoring wet $N_r$ deposition:
$WetN_{ij} = \alpha_j + \beta_i \times P_{ij} \times (S_{ABL})_{ij} + \varepsilon_{ij}$  (9)
$S_{ABL} = S_{trop} \times \dfrac{\int_0^{ABL} \rho(Z)dx}{\int_0^{h_{trop}} \rho(Z)dx}$  (10)
$WetN_{ij}$ is wet $NO_3^-$-N or $NH_4^+$-N deposition at month i and site j; $(S_{ABL})_{ij}$ is the
atmospheric boundary layer (ABL) $NO_2$ or $NH_3$ columns at month i and site j; $P_{ij}$ is
precipitation at month i and site j; $\beta_i$ and $\alpha_j$ are the slope and intercept of random
effects, representing seasonal variability and spatial effects;$\varepsilon_{ij}$ represents the random
error at month i and site j.
The scavenging process of wet $N_r$ deposition usually starts from the height of rainfall
rather than the top of the troposphere, so it is more reasonable to use $NO_2$ and $NH_3$
column below the height of rainfall to build the wet $N_r$ deposition model. The $NO_2$
and $NH_3$ column within ABL is used to build the wet deposition model since
precipitation height is close to the height of the ABL (generally less than 2-3 km).



## 4. Satellite-derived Surface $N_r$ Concentration and Deposition

### 4.1 Surface $NO_2$ Concentration and Oxidized $N_r$ Deposition

The spatial resolutions of global ACTMs and therefore modeled surface $N_r$ concentration are very coarse (for example, the spatial resolution of the global version of GEOS-Chem is $2^o \times 2.5^o$). Thus it can be hard to estimate surface $N_r$ concentration and deposition at a fine resolution at a global scale by ACTMs alone. Instead, the satellite $N_r$ retrievals have a high spatial resolution and can reveal more spatial details than ACTM simulations.

Cheng et al. (Cheng et al., 2013) and Jia et al. (Jia et al., 2016) established a linear model between the surface $NO_2$ concentration and $NO_2$ column by assuming the ratio of the surface $NO_2$ concentration to the tropospheric $NO_2$ column to be fixed, and then used the linear model to convert satellite $NO_2$ columns to surface $NO_2$ concentration, and finally estimated dry $NO_2$ deposition using the inferential method. However, these statistical methods by Cheng et al. (Cheng et al., 2013) and Jia et al. (Jia et al., 2016) are highly dependent on the ground-based measurements, and the established linear models may be not effective over regions with few monitoring sites. A comprehensive study (Nowlan et al., 2014) estimated global surface $NO_2$ concentration during 2005-2007 by multiplying OMI tropospheric $NO_2$ columns by the ACTM-modeled ratio between surface $NO_2$ concentration and tropospheric column (**Fig. 2**). Nowlan et al. (Nowlan et al., 2014) also estimated dry $NO_2$ deposition using the OMI-derived surface $NO_2$ concentration combining the modeled $V_d$ during 2005-2007. This approach followed an earlier study (Lamsal et al., 2008), that focus on North America. As reported by Lamsal et al. (Lamsal et al., 2008), the satellite-derived surface $NO_2$ concentration was generally lower than ground-based $NO_2$ observations, ranging from -17% to -36% in North America. Kharol et al.





(Kharol et al., 2015) used a similar method and found the satellite-derived surface
NO$_2$ concentration was only half of the ground-measured values in North America
(Kharol et al., 2015).

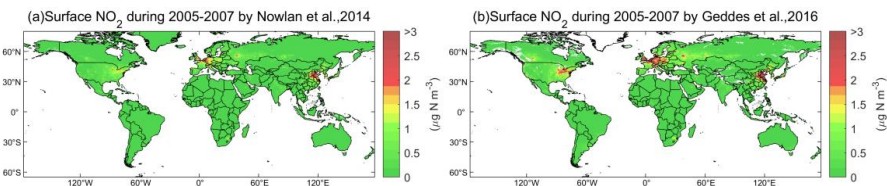

**Fig. 2** Satellite-derived surface NO$_2$ concentration during 2005-2007 by Nowlan et al. (Nowlan et
al., 2014) (a) and by Geddes et al. (Geddes et al., 2016) (b). We gained the surface NO$_2$
concentration by Nowlan et al. (Nowlan et al., 2014) and by Geddes et al. (Geddes et al., 2016) at
the web: http://fizz.phys.dal.ca/~atmos/martin/?page_id=232.

Geddes et al. (Geddes et al., 2016) followed previous studies, and used NO$_2$ column
from the GOME, SCIAMACHY, and GOME-2 to estimate surface NO$_2$ concentration.
Although Geddes et al. (Geddes et al., 2016) did not evaluate their results with
ground-based observation, it is obvious that their surface NO$_2$ estimates were higher
than Nowlan's estimates (Nowlan et al., 2014) based on OMI (**Fig. 2**). This may be
because the OMI-derived NO$_2$ column is much lower than that derived by GOME,
SCIAMACHY, and GOME-2, especially over polluted regions. For example, in China,
the OMI NO$_2$ column is about 30% lower than that of SCIAMACHY and GOME-2
consistently (**Fig. 3**).

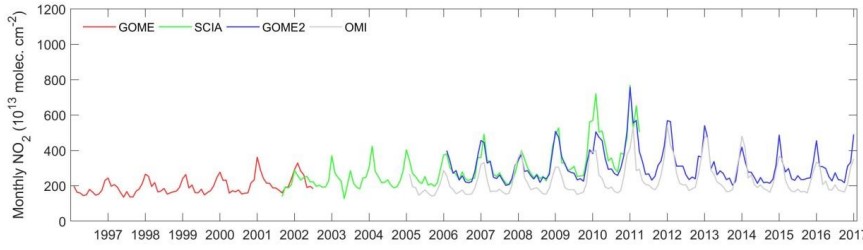

**Fig. 3** An example of the time series of monthly NO$_2$ column retrieved by GOME, SCIAMACHY,
GOME2 and OMI in China. We obtained the GOME, SCIAMACHY, GOME2 and OMI data from
http://www.temis.nl/airpollution/no2.html.

Larkin et al. (Larkin et al., 2017) established a land-use regression model to estimate



global surface $NO_2$ concentration by combining satellite-derived surface $NO_2$
concentration by Geddes et al. (Geddes et al., 2016) and ground-based annual $NO_2$
measurements. The study by Larkin et al. (Larkin et al., 2017) can be considered as
using the ground-based annual measurements to adjust the satellite-derived surface
$NO_2$ concentration by Geddes et al. (Geddes et al., 2016), which helped reduce the
discrepancy between satellite-derived and ground-measured $NO_2$ concentration. The
regression model captured 54% of global $NO_2$ variation, with an absolute error of 2.32
$\mu g\ N\ m^{-3}$.
Zhang et al. (Zhang et al., 2017) followed the framework in **Sect. 3** to estimate the
OMI-derived surface $NO_2$ concentration (at ~50 m) in China, and found good
agreement with ground-based surface $NO_2$ concentration from the NNDMN at yearly
scale (slope=1.00, $R^2$=0.89). The methods by Zhang et al. (Zhang et al., 2017) can
also generate OMI-derived $NO_2$ concentration at any height by the constructed $NO_2$
vertical profile. Zhang et al. (Zhang et al., 2017) also estimated dry $NO_2$ deposition
using the OMI-derived surface $NO_2$ concentration combining the modeled $V_d$ during
2005-2016. Based on Zhang's estimates, the Gaussian function can well simulate the
vertical distribution of $NO_2$ from an ACTM (MOZART) (Emmons et al., 2010) with
99.64% of the grids having $R^2$ values higher than 0.99. This suggests that the
ACTM-simulated vertical distribution of $NO_2$ has a general pattern, which can be
emulated by Gaussian functions. Once a vertical profile was constructed, it can be
easily used to estimate $NO_2$ concentration at any height.
In this study, we used the framework in **Sect. 3** to estimate the OMI-derived surface
$NO_2$ concentration globally. To validate the OMI-derived surface $NO_2$ concentrations,
ground-measured surface $NO_2$ concentration in China, the US and Europe in 2014
was collected (**Fig. 4**). The total number of $NO_2$ observations in China, the US and


Europe are 43, 373 and 88 respectively. The OMI-derived annual average for all sites
was 3.74 μg N m$^{-3}$, which was close to the measured average (3.06 μg N m$^{-3}$). The R$^2$
between OMI-derived surface NO$_2$ concentrations and ground-based NO$_2$
measurements was 0.75 and the RMSE was 1.23 μg N m$^{-3}$ (**Fig. 5**), which is better
than the modeling results by the GEOS-Chem ACTM (R$^2$=0.43, RMSE=1.93 μg N
m$^{-3}$). Satellite-based methods have the advantages of spatiotemporally continuous
monitoring N$_r$ at a higher resolution, which helps alleviate the problem of the coarse
resolution of ACTMs in estimating N$_r$ concentration and deposition.

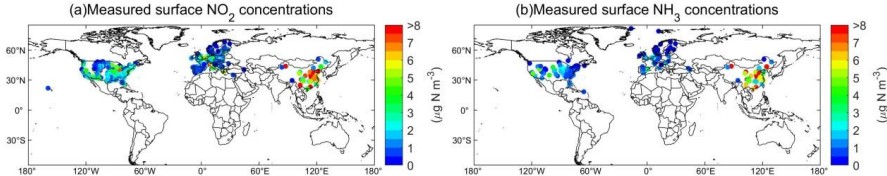


**Fig. 4** Spatial distribution of measured surface NO$_2$ and NH$_3$ concentrations in 2014. For NO$_2$ (a),
the measured data in China, the US and Europe were obtained from the NNDMN, US-EPA and
EMEP, respectively; for NH$_3$ (b), the measured data in China, the US and Europe were obtained
from the NNDMN, US-AMoN and EMEP, respectively

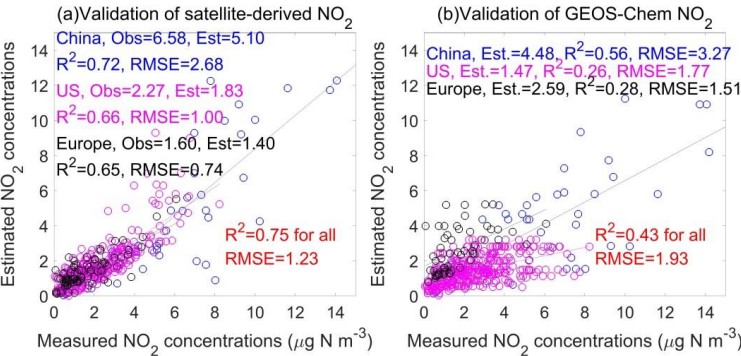


**Fig. 5** Comparison between annual mean satellite-derived and ground-measured surface NO$_2$
concentrations (a), and comparison between annual mean modeled (by an ACTM as GEOS-Chem)
and ground-measured surface NO$_2$ concentrations (b). The ground-based monitoring sites are
shown in **Fig. 4**.

For NO$_3^-$ and HNO$_3$, previous studies firstly constructed the relationship between NO$_2$,
NO$_3^-$ and HNO$_3$, and found a relatively high linear relationship between NO$_2$, NO$_3^-$,





and $HNO_3$ at a monthly or yearly scale. For example, Jia et al. (Jia et al., 2016) found
a linear relationship between $NO_2$ and $NO_3^-$, $HNO_3$ concentration at annual scale
($R^2$=0.70). Similarly, based on the ground-based measurements in the NNDMN, a
high correlation was found between surface $NO_2$ and $NO_3^-$ concentration at monthly
or annual timescales (**Fig. 6**) (Liu et al., 2017c). Using these linear relationships and
satellite-derived surface $NO_2$ concentration, the annual mean surface $NO_3^-$ and $HNO_3$
can be estimated. Alternatively, the relationship of $NO_2$, $NO_3^-$ and $HNO_3$ can also be
modeled by an ACTM. For example, a strong relationship of tropospheric $NO_2$, $NO_3^-$
and $HNO_3$ column was simulated over all months by an ACTM, with the correlation
ranging from 0.69 to 0.91 (Liu et al., 2017a). But, over shorter timescales, the
relationship between $NO_2$, $NO_3^-$ and $HNO_3$ may be nonlinear, which we should be
cautious about when estimating surface $NO_3^-$ and $HNO_3$ concentration from $NO_2$
concentration.



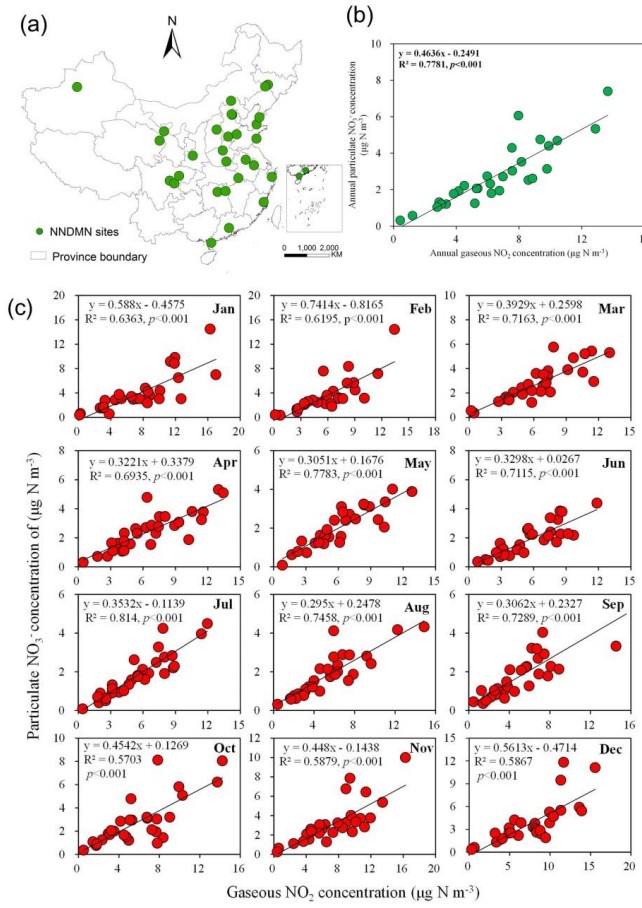

**Fig. 6** Correlation between surface $NO_2$ and particulate $NO_3^-$ concentration in the NNDMN at annual and monthly scales, which were adopted from Liu et al. (Liu et al., 2017c). (a) indicates the spatial locations of monitoring sites in the NNDMN; (b) and (c) represent yearly and monthly relationship between surface $NO_2$ and particulate $NO_3^-$ concentration, respectively.

For the wet $N_r$ deposition, Liu et al. (Liu et al., 2017a) followed the framework in

**Sect. 3** to estimate wet nitrate deposition using ABL $NO_2$ columns derived from OMI

$NO_2$ column and $NO_2$ vertical profile from an ACTM (MOZART), and precipitation

by a mixed-effects model showing the proposed model can achieve high predictive

power for monthly wet nitrate deposition over China (R=0.83, RMSE=0.72).

**4.2 Surface $NH_3$ Concentration and Reduced $N_r$ Deposition**

With the development of atmospheric remote sensing of $NH_3$, some scholars have





estimated surface $NH_3$ concentration and dry $NH_3$ deposition based on the satellite
$NH_3$ column data. Assuming the ratio between the surface $NH_3$ concentration to the
$NH_3$ column was fixed, Yu et al. (Yu et al., 2019) applied a linear model to convert
satellite $NH_3$ columns to surface $NH_3$ concentration and estimated dry $NH_3$ deposition
in China using the inferential method. But Yu et al. (Yu et al., 2019) did not consider
the spatial variability of the vertical profiles of $NH_3$, which may cause a large
uncertainty in estimating surface $NH_3$ concentration.
In Western Europe, Graaf et al. (Graaf et al., 2018) used the ratio of the surface $NH_3$
concentration (in the bottom layer) to total $NH_3$ column from an ACTM to convert the
IASI $NH_3$ column to surface $NH_3$ concentration, and then estimated dry $NH_3$
deposition combining the modeled deposition velocity and IASI-derived surface $NH_3$
concentration. Similarly, in North America, Kharol et al. (Kharol et al., 2018)
estimated the dry $NH_3$ deposition by the CrIS-derived surface $NH_3$ concentration and
deposition velocity of $NH_3$. They found a relatively high correlation (R=0.76)
between the CrIS-derived surface $NH_3$ concentration and AMoN measurements during
warm seasons (from April to September) in 2013 (**Fig. 7**). Over China, Liu et al. (Liu
et al., 2017b) found a higher correlation (R=0.81) between IASI-derived surface $NH_3$
concentrations and the measured surface $NH_3$ concentrations than those from an
ACTM (R=0.57, **Fig. 8**).

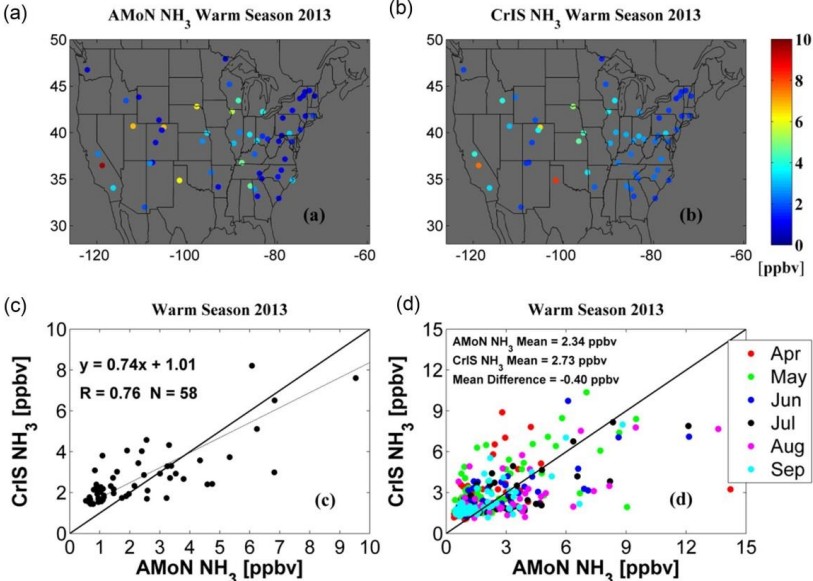


**Fig. 7** Comparisons of the measured surface $NH_3$ concentration by the AMoN and CrIS-derived
surface $NH_3$ concentration in the US during warm season (April-September) in 2013 (Kharol et al.,
2018). (a) and (b) indicate measured and CrIS-derived surface $NH_3$ concentration at the AMoN
sites, respectively; (c) represents the comparison of averaged surface $NH_3$ concentration during
warm months between CrIS-derived estimates and measurements, while (d) indicates the
comparison of monthly surface $NH_3$ concentration between CrIS-derived estimates and
measurements.

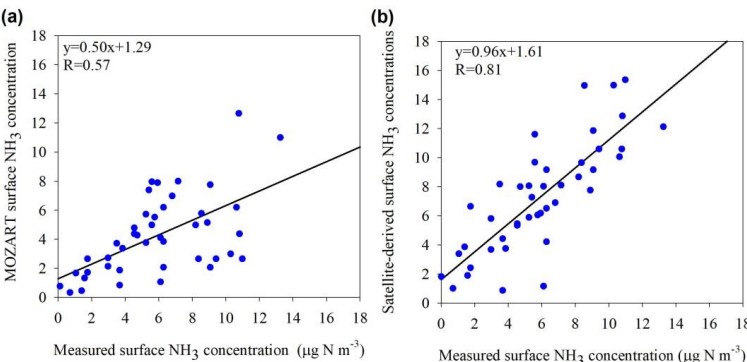

517

**Fig. 8** Comparisons of the measured surface $NH_3$ concentration with IASI-derived surface $NH_3$
concentration at the NNDMN sites over China (Liu et al., 2017b). (a) indicates the comparison of
measured and modeled surface $NH_3$ concentration from an ACTM (MOZART), and (b) represents
the comparison of the measured and IASI-derived surface $NH_3$ concentration.

Liu et al. (Liu et al., 2019) followed the framework in **Sect. 3** to estimate the

IASI-derived surface $NH_3$ concentration (at the middle height of the first layer by an





ACTM) (**Fig. 9**), and found a good agreement with ground-based surface $NH_3$
concentration. The correlation between the measured and satellite-derived annual
mean surface $NH_3$ concentrations over all sites was 0.87 as shown in **Fig. 10**, while
the average satellite-derived and ground-measured surface $NH_3$ concentration was
2.52 and 2.51 µg N m$^{-3}$ in 2014 at the monitoring sites, respectively. The
satellite-derived estimates achieved a better accuracy ($R^2$=0.76, RMSE = 1.50 µg N
m$^{-3}$) than an ACTM (GEOS-Chem, $R^2$=0.54, RMSE = 2.14 µg N m$^{-3}$).

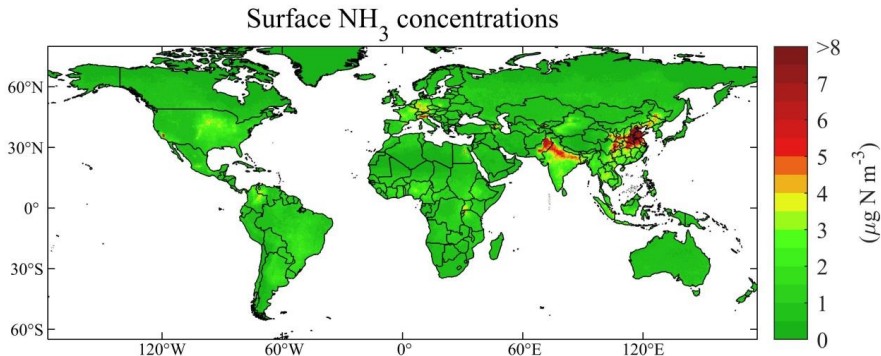


**Fig. 9** Spatially satellite-based surface $NH_3$ estimates in 2014 (Liu et al., 2019). The global surface
$NH_3$ concentration datasets have been released on the website:
https://zenodo.org/record/3546517#.Xj6I4GgzY2w.

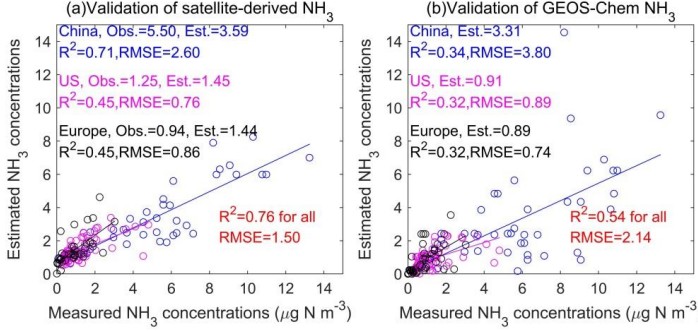


**Fig. 10** Comparison between yearly satellite-based and measured surface $NH_3$ concentrations (a),
and comparison between yearly modeling (by an ACTM as GEOS-Chem) and measured surface
$NH_3$ concentrations (b) (Liu et al., 2019). The ground-based monitoring sites are shown in **Fig. 4**.

The proposed methods (Liu et al., 2019) can also estimate $NH_3$ concentration at any





height using the constructed vertical profile function of $NH_3$. The Gaussian function
can well emulate the vertical distribution of $NH_3$ from an ACTM outputs with 99% of
the grids having $R^2$ values higher than 0.90 (**Fig. 11**). This means, for regional and
global estimation, the vertical distribution of $NH_3$ concentration has a general pattern,
which can be mostly emulated by the Gaussian function. Once a global $NH_3$ vertical
profile was simulated, it can be easily used to estimate satellite-derived $NH_3$
concentration at any height. We can also estimate dry $NH_3$ deposition using the
IASI-derived surface $NH_3$ concentration combining the modeled $V_d$. To date, there are
still no studies developing satellite-based methods to estimate the wet reduced $N_r$
deposition on a regional scale.

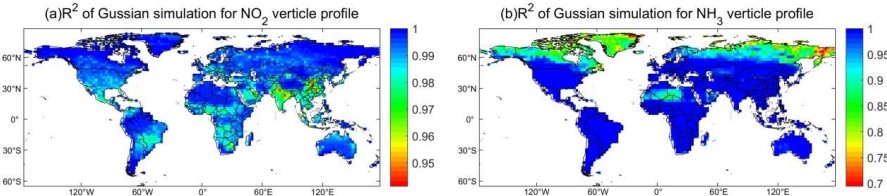


**Fig. 11** Spatial distributions of $R^2$ for Gaussian function by simulating $NH_3$ and $NO_2$ vertical
profiles. This is an example of Gaussian fitting using 47 layers' $NH_3$ and $NO_2$ concentration from
an ACTM (GEOS-Chem).

**5. Trends of Surface $N_r$ Concentration and Deposition by Satellite-based**
**Methods**
The $N_r$ concentration and deposition modeled by ACTMs are highly dependent on the
accuracy of input $N_r$ emissions. The methods commonly used to estimate
anthropogenic $N_r$ emissions are based on the data of human activities and emission
factors, which can be highly uncertain. The ACTM methods driven by $N_r$ emission
inventory have relatively poor timeliness, and have limitations in monitoring the
recent trends of $N_r$ deposition.
Satellite-based methods provide a simple, fast and relatively objective way to
monitoring $N_r$ deposition at a high resolution, and less susceptible to the errors in the





assumptions that emission inventories are compiled based on, particularly the lack of
reliable data over developing countries (Crippa et al., 2018). With such advantages,
researchers developed the satellite-based methods to estimate surface $N_r$ concentration,
deposition and even emissions. Satellite-based methods have advantages in
monitoring the recent trends of $N_r$ deposition. Geddes et al. (Geddes and Martin, 2017)
used $NO_2$ column from the GOME, SCIAMACHY, and GOME-2 to estimate
satellite-derived $NO_x$ emissions, and then used the calibrated $NO_x$ emission inventory
to drive an ACTM to simulate the long-term oxidized $N_r$ deposition globally. They
found oxidized $N_r$ deposition from 1996 to 2014 decreased by 60% in Eastern US,
doubled in East China, and declined by 20% in Western Europe (**Fig. 12**). We use the
datasets by Geddes et al. (Geddes and Martin, 2017) to calculate the trends of total
oxidized $N_r$ deposition during 1996-2014. It is obvious that two completely opposite
trends exist: (1) in East China with a steep increase of higher than 0.5 kg N $ha^{-1}$ $y^{-1}$
and (2) East US with a steep decrease of lower than -0.5 kg N $ha^{-1}$ $y^{-1}$. Although it is
not a direct way to use satellite $N_r$ observation to estimate $N_r$ deposition, the method
of estimating trends of $N_r$ deposition by Geddes et al. (Geddes and Martin, 2017) can
be considered effective since it took account of the changes of both $NO_x$ emission and
climate by an ACTM.

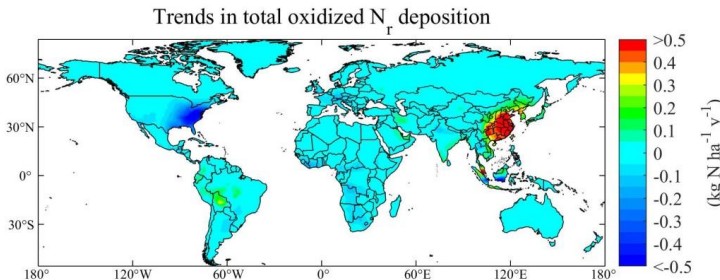


**Fig. 12** Gridded annual changes of total oxidized $N_r$ deposition simulated by GEOS-Chem
constrained with GOME, SCIAMACHY, and GOME-2 $NO_2$ retrievals during 1996-2014 (Geddes
and Martin, 2017). We gained the generated datasets



(http://fizz.phys.dal.ca/~atmos/martin/?page_id=1520) by Geddes et al. (Geddes and Martin,
591                     2017), and calculated the trends using the linear methods.

Some researchers developed a more direct way to infer the trends of surface $N_r$
concentration and deposition. Geddes et al. (Geddes et al., 2016) presented a
comprehensive long-term global surface $NO_2$ concentration estimate (at 0.1°
resolution using an oversampling approach) between 1996 and 2012 by using $NO_2$
column from the GOME, SCIAMACHY, and GOME-2. The surface $NO_2$
concentration in North America (the US and Canada) decreased steeply, followed by
Western Europe, Japan and South Korea, while approximately tripled in China and
North Korea (Geddes et al., 2016). Jia et al. (Jia et al., 2016) established a simple
linear regression model based on OMI $NO_2$ column and ground-based surface $N_r$
concentration, and then estimated the trends of dry $N_r$ deposition globally between
2005 and 2014. They found that dry $N_r$ deposition in Eastern China increased rapidly,
while in the Eastern US, Western Europe, and Japan dry $N_r$ deposition has decreased
in recent decades.
We split the time span of 2005-2016 into two periods: 2005-2011 and 2011-2016, as
surface $NO_2$ concentration shows opposite trend in China in these two periods. The
magnitudes of both growth and decline in surface $NO_2$ concentration in China are
most pronounced worldwide in the two periods (**Fig. 13**). During 2005-2011, apart
from Eastern China with the largest increase in surface $NO_2$ concentration, there are
also several areas with increasing trends such as Northwest and East India (New Delhi
and Orissa), Western Russia, Eastern Europe (Northern Italy), Western US (Colorado
and Utah), Northwestern US (Seattle and Portland), Southwestern Canada (Vancouver,
Edmonton, Calgary), Northeast Pakistan and Northwest Xinjiang (Urumqi). Notably,
the biggest decreases in surface $NO_2$ concentration during 2005-2011 occurred in
Eastern US and Western EU (North France, South England, and West German).





During 2011-2016, due to the strict control of $NO_x$ emissions, Eastern China had the

largest decrease in surface $NO_2$ concentration than elsewhere worldwide, followed by

Western Xinjiang, Western Europe and some areas in Western Russia.

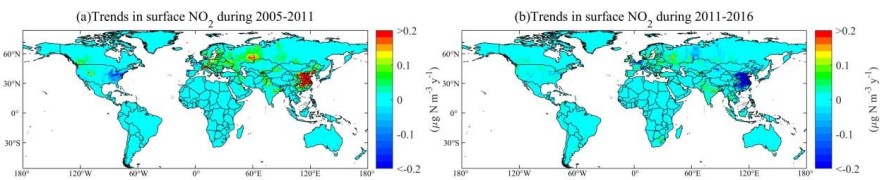

**Fig. 13** Gridded annual changes in surface $NO_2$ concentrations gained by OMI retrievals during
2005-2011 (a) and during 2011-2016 (b) in this study. We have released the global surface $NO_2$
concentrations during 2005-2016 available at the website:
https://zenodo.org/record/3546517#.Xj6I4GgzY2w.

Liu et al. (Liu et al., 2019) estimated surface $NH_3$ concentration globally during

2008-2016 using satellite $NH_3$ retrievals by IASI. A large increase of surface $NH_3$

concentrations was found in Eastern China, followed by Northern Xinjiang province

in China during 2008-2016 (**Fig. 14**). Satellite-based methods have been proven as an

effective and unique way to monitoring the trends of global $N_r$ concentration and

deposition. To date, there are still few studies reporting the satellite-derived trends of

reduced $N_r$ deposition on a global scale.

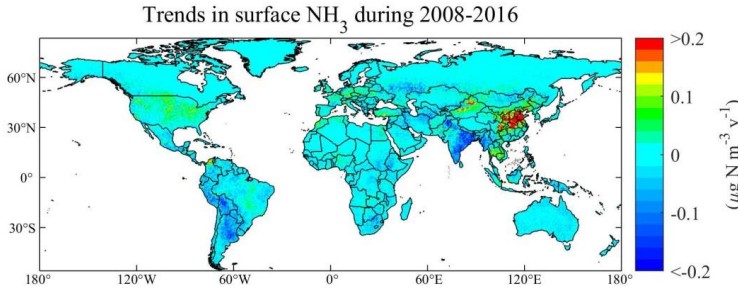

**Fig. 14** Gridded annual changes in surface $NH_3$ concentrations gained by IASI retrievals during
2008-2016 (Liu et al., 2019). We have released the global surface $NH_3$ concentrations during
2008-2016 at the website: https://zenodo.org/record/3546517#.Xj6I4GgzY2w.





## 6. Remaining Challenges for Estimating $N_r$ Deposition Using Satellite Observation

First, the reduced $N_r$ deposition plays an important contribution to total $N_r$ deposition. $NH_3$ exhibits bi-directional air-surface exchanges. The $NH_3$ compensation point (Farquhar et al., 1980) is also an important and highly variable factor controlling dry $NH_3$ deposition (Schrader et al., 2016;Zhang et al., 2010). However, the current existing satellite-based methods did not consider this bi-directional air-surface exchange. It is important to better parameterize the $NH_3$ compensation point, and assess the effects of bi-directional air-surface exchanges on estimating the dry $NH_3$ deposition.

Second, the existing satellite-based methods to estimate $N_r$ deposition used the ratio of the surface $N_r$ concentration to the $N_r$ column by an ACTM to convert satellite $N_r$ column to surface $N_r$ concentration. However, the calculated ratio (by an ACTM) and the satellite $N_r$ column have different spatial resolutions, and previous studies usually applied the modeled ratio directly or interpolate the ratio into the resolution of satellite $N_r$ column. This method assumes the relationship at coarse resolution by an ACTM can also be effective in fine resolution as satellite indicated. When regional studies are conducted, regional ACTMs coupled with another meteorological model (e.g. WRF-Chem, WRF-CMAQ) (Grell et al., 2005;Wong et al., 2012) can be configured to match the spatial resolution of satellite observation, but this is not as viable for global ACTMs (e.g. MOZART, GEOS-Chem) due to differences in model structures and computational cost. The modeled ratio of surface $N_r$ concentration to the $N_r$ column may have variability at spatial scales finer than the horizontal resolution of global ACTMs. The impact of such scale effect (at different spatial scales) on estimated surface $N_r$ concentration should be further studied.



Third, the satellite observation can only obtain reliable $NO_2$ and $NH_3$ column
presently, and there are no available high-resolution and reliable direct $HNO_3$, $NO_3^-$,
$NH_4^+$ retrievals. For $HNO_3$, $NO_3^-$, $NH_4^+$ concentrations, the satellite-based methods
often applied the satellite-derived $NO_2$ and $NH_3$ concentration and the relationship
between $N_r$ species from an ACTM (or ground-based measurements) to estimate
surface $HNO_3$, $NO_3^-$, $NH_4^+$ concentration. With the development of satellite
technology, more and more $N_r$ species can be detected, such as $HNO_3$. However, at
present, satellite $HNO_3$ products are not mature, and the spatial resolution is low.
Direct, high-resolution and reliable satellite monitoring of more $N_r$ species is critical
to further developing the use of using atmospheric remote sensing to estimate $N_r$
deposition at global and regional scales.
Fourth, estimating wet $N_r$ deposition using satellite $NO_2$ and $NH_3$ column remains
relatively uncommon. Further studies should focus on how to combine the
high-resolution satellite $NO_2$ and $NH_3$ column and the ground-based monitoring data
to build wet $N_r$ deposition models to estimate wet $N_r$ deposition at higher
spatiotemporal resolution. The proposed scheme to estimate the wet $N_r$ deposition in
**Sect. 3** is statistical. On the other hand, the wet $N_r$ deposition includes the scavenging
processes of in-cloud, under-cloud and precipitation. Processed-level knowledge and
models can benefit the estimation of wet $N_r$ deposition using satellite $NO_2$ and $NH_3$
column.
**7. Conclusion**
The recent advances of satellite-based methods for estimating surface $N_r$
concentration and deposition have been reviewed. Previous studies have focused on
using satellite $NO_2$ column to estimate surface $NO_2$ concentrations and dry $NO_2$
deposition both regionally and globally. The research on calculating surface $NH_3$



concentration and reduced $N_r$ deposition by satellite $NH_3$ data is just beginning, and
some scholars have carried out estimating surface $NH_3$ concentration and dry $NH_3$
deposition on different spatial and temporal scales, but the research degree is still
relatively low. We present a framework of using satellite $NO_2$ and $NH_3$ column to
estimate $N_r$ deposition based on recent advances. The proposed framework of using
Gaussian function to model vertical $NO_2$ and $NH_3$ profiles can be used to convert the
satellite $NO_2$ and $NH_3$ column to surface $NO_2$ and $NH_3$ concentration at any height
simply and quickly. The proposed framework of using satellite $NO_2$ and $NH_3$ column
to estimate wet $N_r$ deposition is a statistical way, and further studies should be done
from a mechanism perspective. Finally, we summarized current challenges of using
satellite $NO_2$ and $NH_3$ column to estimate surface $N_r$ concentration and deposition
including a lack of considering $NH_3$ bidirectional air-surface exchanges and the
problem of different spatial scales between an ACTM and satellite observation.
**Acknowledgments**
This study is supported by the National Natural Science Foundation of China (No.
41471343, 41425007 and 41101315) and the Chinese National Programs on Heavy
Air Pollution Mechanisms and Enhanced Prevention Measures (Project No. 8 in the
2nd Special Program).
**Author contributions**. LL designed this study. LL, YYY and WX conducted the data
analysis. All co-authors contributed to the revision of the paper.
**Data availability**. OMI $NO_2$ datasets are available at
http://www.temis.nl/airpollution/no2.html. IASI $NH_3$ datasets are available at
https://cds-espri.ipsl.upmc.fr/etherTypo/index.php?id=1700&L=1. Surface $NO_2$
concentration during 2005-2007 obtained by Nowlan et al. (Nowlan et al., 2014) and
longterm estimates (1996-2012) by Geddes et al. (Geddes et al., 2016) are available at



http://fizz.phys.dal.ca/~atmos/martin/?page_id=232. Total oxidized $N_r$ deposition
simulated by GEOS-Chem constrained with GOME, SCIAMACHY, and GOME-2
$NO_2$ retrievals during 1996-2014 (Geddes and Martin, 2017) is available at
http://fizz.phys.dal.ca/~atmos/martin/?page_id=1520. A database of atmospheric $N_r$
concentration and deposition from the nationwide monitoring network in China is
available at https://www.nature.com/articles/s41597-019-0061-2. Measured $N_r$
concentration and deposition datasets in the United States are available on the website:
https://www.epa.gov/outdoor-air-quality-data. Measured surface $NO_2$ and $NH_3$
concentration datasets in Europe are available at
https://www.nilu.no/projects/ccc/emepdata.html. Global surface $NO_2$ and $NH_3$
concentration data used to calculate the longterm trends in **Fig. 13** and **Fig. 14** have
been released on the website: https://zenodo.org/record/3546517#.Xj6I4GgzY2w.
**Competing interests**. The authors declare no competing financial interests.

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
