# Peer review of "Reviewing Global Estimates of Surface Reactive Nitrogen Concentration and Deposition Using Satellite Observation Lei Liu [a, *], Xiuying Zhang [b], Wen Xu [c], Xuejun Liu [c], Xuehe Lu [b], Jing Wei [d, e], Yi Li [f],"

_Atmospheric Chemistry and Physics, 2020_

## Referee Comment (RC1) · Anonymous Referee #1 · 17 Apr 2020

This study reviews recent literatures on estimating reactive nitrogen (Nr) deposition using the satellite retrievals of $NO_2$ and $NH_3$, proposes a framework of using satellite data to estimate Nr deposition, and suggests a few research challenges. The topic of nitrogen deposition is important, and the compilation of recent literatures on reactive nitrogen deposition is useful to the research community. However, the manuscript mainly gives general descriptions of the previous results but lacks critical analysis and synthesis. The uncertainties in satellite measurements and chemical transport models, which are key to estimating Nr deposition based on satellite column measurements, are not addressed in detail. Overall, the scientific values of this work could be enhanced by more in-depth discussion of the advancement, challenges, and directions for future research.

**Specific comments:**

1. The authors highlight the advantages of satellite-based method compared to ground-based monitoring and ACTM simulation method. But there are significant uncertainties of satellite column measurements, especially for NH3. In addition, the satellite-based method strongly depends on the ACTM simulation. What are the key uncertainties of the ACTM related to deposition estimates? How do the uncertainties in satellite measurements and ACTM affect satellite-based estimation? What are recommendations to reduce these uncertainties?

2.The authors propose a framework for combining satellite data, ground-based monitoring and ACTM (Figure 1). But it is not clear if it is a new idea. It seems that the approach has already been used in previous studies as indicated in the literatures shown in sections after Figure 1.

3. The title contains "Nr concentration and deposition", but the introduction part and the framework only mentions "deposition". In my opinion, the estimation of Nr concentrations is just a part of the estimation of Nr depositions. There are many other studies which have offered more in-depth discussions of column concentrations of $NO_2$ and $NH_3$. I am not saying that concentrations cannot be shown but suggest framing the paper with a clearer focus on deposition.

4. Line 193-195: Why may this method lead to an underestimation of surface $NO_2$ concentration? In your proposed framework, the similar method has been used to estimate the surface $NO_2$ concentration. Why is there no large underestimation in your validation? While you use the Gaussian function to fit the vertical concentration profile, but for the surface layer, you still use the ACTM-derived the relationship between the $NO_2$ column and surface $NO_2$ concentration.

5. Line 405-409: The derived $NO_2$ columns from these satellites are quite different. Can you give some suggestions to the readers about which satellite data to use? Why do you choose OMI $NO_2$ in your estimation? What are the results if you use other satellite data?

6. Line 550-552: Can the similar method in equation 9 and 10 be used to estimate wet reduced Nr depositions? What are the different challenges for the estimations of wet reduced Nr depositions, compared with oxidized Nr?

7. Section 5: For the trend estimation of Nr concentrations and depositions, have you conducted

ACTM simulation for each year? The changes in emission and meteorology can significantly affect the Nr vertical profile and Nr species ratio, which are important in your satellite-based estimation.

8. Line 567-569: This statement needs to be modified. As mentioned above, the satellite-based method strongly depends on the ACTM simulation. The uncertainties in emission inventories and other parts of ACTM can also significantly affect the vertical distribution of pollutants and the ratios of $NO_2$ and other Nr species (e.g. $HNO_3$, $NH_4^+$).

9. Line 697: Are there any previous studies using a mechanism method to estimate Nr deposition?

**Minor comments:**

1. The authors should give the definition of reactive nitrogen (Nr). "Nr (such as $NO_3^-$ and $NH_4^+$)" is mentioned in line 48, and "Nr (NOx and $NH_3$)" is mentioned in line 59. This is confusing.
2. Line 57, change "mineral energy" to "fossil energy".
3. Line 83, add "and" between the two words "accurate quantitative".
4. Line 145-146: "Tian et al." should be "Tan et al. (2018)".
5. Line 170: "Cheng et al. (Cheng et al., 2013)" should be "Cheng et al. (2013)". Please check the citation format throughout the manuscript.
6. Line 170-171: This sentence is not easy to understand. Please revise it.
7. Line 198-200: The study of Larkin et al., 2017 should be put in the previous paragraph discussing the method using the satellite data and statistical model. I think that the authors are discussing the method using the satellite data and ACTM-derived relationship in this paragraph.
8. Line 225-232: This information based on Jia et al. (2016) has been mentioned in line 176-184. They are repetitive.

---

## Referee Comment (RC2) · Anonymous Referee #2 · 1 Jun 2020

This manuscript discusses recent advances of estimating surface Nr concentration and deposition, presents a framework of using satellite data to estimate surface Nr concentration and deposition, and summarizes the existing challenges for the satellite-based methods.

The manuscript is very clearly written and logically organized. It provides sufficient and up-to-date literature citations. Listed below comments and suggestions for changes are relatively minor, but should be carefully considered. I recommend publication after addressing following comments:

1. L290: It is unclear to me how the vertical resolution of GEOS-Chem can resolve the
vertical gradients that are likely to exist in source regions. The authors should clarify these several issues: (1) the vertical structure of the model, (2) the measurement characteristics of the surface observation (including height), (3) how this information is used to calculate surface concentrations. 2. Fig. 10b: It is true that NH3 can be more accurately retrieved in one region than another depending on the thermal contrast. But it is not clear to me why this would be so much better in China than that in the US? I guess it is also just a matter of detection limits? It could also be related to more reliable simulation of mixing, depending on sufficient observational input into the parent weather model. Please clarify this issue. 3. L531: For the estimated ammonia deposition, its uncertainties from remote sensing and models should be discussed more in this manuscript. 4. title: I suggest to change the satellite observation to "satellite retrievals" since IASI NH3 data were not a direct satellite observation but a reanalysis data using the statistical methods. 5. L30: The abbreviation must be defined for the first occurrence. 6. L137: Replace ACTM with Atmospheric chemistry transport model 7. L306: Added the references of the equations. 8. L333: Added the references of the equations.

Please also note the supplement to this comment:
https://www.atmos-chem-phys-discuss.net/acp-2020-91/acp-2020-91-RC2-supplement.pdf

---

## Author Comment (AC1) · 3 Jun 2020

**Response to Referee #1**

This study reviews recent literatures on estimating reactive nitrogen (Nr) deposition using the satellite retrievals of NO2 and NH3, proposes a framework of using satellite data to estimate Nr deposition, and suggests a few research challenges. The topic of nitrogen deposition is important, and the compilation of recent literatures on reactive nitrogen deposition is useful to the research community. However, the manuscript mainly gives general descriptions of the previous results but lacks critical analysis and synthesis. The uncertainties in satellite measurements and chemical transport models, which are key to estimating Nr deposition based on satellite column measurements, are not addressed in detail. Overall, the scientific values of this work could be enhanced by more in-depth discussion of the advancement, challenges, and directions for future research.

The authors appreciate the valuable suggestions given by Referee #1 for improving the overall quality of the manuscript. In this document, we describe how we addressed the reviewer's comments. Detailed responses to each comment are given below (in blue).

**Specific comments:**

1. The authors highlight the advantages of satellite-based method compared to ground-based monitoring and ACTM simulation method. But there are significant uncertainties of satellite column measurements, especially for $NH_3$. In addition, the satellite-based method strongly depends on the ACTM simulation. What are the key uncertainties of the ACTM related to deposition estimates? How do the uncertainties in satellite measurements and ACTM affect satellite-based estimation? What are recommendations to reduce these uncertainties?

Yes, the uncertainties mainly came from the satellite retrievals and ACTM simulation. We did not aim to improve the accuracy of the satellite observations or the ACTM themselves, but to combine their advantages to gain surface $N_r$ concentrations with better performance with the ground-based measurements. We have added the following text for more clarifications in the discussion:

"For the dry deposition, the uncertainty mainly came from the satellite-derived estimates using the modeled vertical profiles. The uncertainty of vertical profiles modeled by CTM mainly resulted from the chemical and transport mechanisms. We recommend using the Gaussian function to determine the height of surface $NO_2$ and $NH_3$ concentrations that best matched with the ground-based measurements. There may exist systematic biases by simply using the relationship of $NO_2$ columns and surface concentration to estimate satellite surface $NO_2$ concentrations."

2. The authors propose a framework for combining satellite data, ground-based monitoring and ACTM (Figure 1). But it is not clear if it is a new idea. It seems that the approach has already been used in previous studies as indicated in the literatures shown in sections after Figure 1.

Yes, it's a new framework proposed in this study. Previous studies mainly focused on the methods to estimate surface $NO_2$ concentrations, while **Fig. 1** shows the general approach for estimating all $N_r$ spices on both concentration and deposition.

3. The title contains "Nr concentration and deposition", but the introduction part and the framework only mention "deposition". In my opinion, the estimation of Nr concentrations is just a part of the estimation of Nr depositions. There are many other studies which have offered more in-depth discussions of column concentrations of NO2 and NH3. I am not saying that concentrations cannot be shown but suggest framing the paper with a clearer focus on deposition.

Thanks for your suggestion. But, we think the introduction is appropriate since the estimation of $N_r$ concentrations is just a part of the estimation of dry $N_r$ depositions. The title included both the "$N_r$ concentration" and "deposition" because we reviewed on the methods of estimating both surface $N_r$ concentration and $N_r$ deposition.

4. Line 193-195: Why may this method lead to an underestimation of surface NO2 concentration? In your proposed framework, the similar method has been used to estimate the surface NO2 concentration. Why is there no large underestimation in your validation? While you use the Gaussian function to fit the vertical concentration profile, but for the surface layer, you still use the ACTM derived the relationship between the NO2 column and surface NO2 concentration.

No, the methods in this study were different from the previous studies. We did not simply use the relationship between the $NO_2$ column and surface $NO_2$ concentration from the CTM. As presented in the main text, we can estimate surface $NO_2$ concentration at any height by using the Gaussian function. We used the surface $NO_2$ concentration at a certain height which best matched with the ground-based measurements.

5. Line 405-409: The derived NO2 columns from these satellites are quite different. Can you give some suggestions to the readers about which satellite data to use? Why do you choose OMI NO2 in your estimation? What are the results if you use other satellite data?

The readers can use any satellite data combining the Gaussian function to estimate surface $NO_2$ concentrations. They can use surface $NO_2$ concentrations at a certain height which best matched with the ground-based measurements. The key is not selecting which satellite data we should use, but determining which height of surface $NO_2$ concentrations that better matched with the ground-based measurements by

Gaussian function.

6. Line 550-552: Can the similar method in equation 9 and 10 be used to estimate wet reduced Nr depositions? What are the different challenges for the estimations of wet reduced Nr depositions, compared with oxidized Nr?

Yes, the methods were the same for estimating both oxidized and reduced $N_r$ deposition. We did not identify big difference in the estimations of wet oxidized and reduced $N_r$ depositions.

7. Section 5: For the trend estimation of Nr concentrations and depositions, have you conducted ACTM simulation for each year? The changes in emission and meteorology can significantly affect the Nr vertical profile and Nr species ratio, which are important in your satellite-based estimation.

Yes, we did. Please note that the simulated profile function has a general rule, which can be well simulated by Gaussian function for any year (for our case during 2005-2016). Thus, there is no need to simulate the vertical profile of $NO_2$ and $NH_3$ for each year.

8. Line 567-569: This statement needs to be modified. As mentioned above, the satellite-based method strongly depends on the ACTM simulation. The uncertainties in emission inventories and other parts of ACTM can also significantly affect the vertical distribution of pollutants and the ratios of NO2 and other Nr species (e.g. HNO3, NH4+).

No, the emission inventories should not affect the vertical profiles shapes using Gaussian function, but the transport and chemical mechanism in the CTM may affect the accuracy of the vertical profile distribution. We mean that the satellite-based methods did not need to rely on the accuracy of the statistical emission data.

9. Line 697: Are there any previous studies using a mechanism method to estimate Nr

deposition?

As far as we know, previous studies using satellite $NO_2$ and $NH_3$ column to estimate wet $N_r$ deposition were through a statistical way, and no studies were done from a mechanism perspective.

**Minor comments:**

1. The authors should give the definition of reactive nitrogen (Nr). "Nr (such as NO3- and NH4+)" is mentioned in line 48, and "Nr (NOx and NH3)" is mentioned in line 59. This is confusing.

We have added the following text for clarifications:

"$N_r$ refers to the general term of N-containing substances in atmosphere, plants, soils and fertilizers that are not combined with carbon".

2. Line 57, change "mineral energy" to "fossil energy".

We have revised it as suggested.

3. Line 83, add "and" between the two words "accurate quantitative".

We have revised it as suggested.

4. Line 145-146: "Tian et al." should be "Tan et al. (2018)".

We have revised it as suggested.

5. Line 170: "Cheng et al. (Cheng et al., 2013)" should be "Cheng et al. (2013)". Please check the citation format throughout the manuscript.

We have checked the citation format throughout the manuscript as suggested.

6. Line 170-171: This sentence is not easy to understand. Please revise it.

We have revised it as follows:

"This method used the simple linear model and did not consider the vertical profiles of $NO_2$ (Cheng et al., 2013)"

7. Line 198-200: The study of Larkin et al., 2017 should be put in the previous

paragraph discussing the method using the satellite data and statistical model. I think that the authors are discussing the method using the satellite data and ACTM-derived relationship in this paragraph.

No, Larkin et al. (2017) were also based on the satellite data and ACTM-derived relationship similar to Geddes et al. (2016), and it should be there.

8. Line 225-232: This information based on Jia et al. (2016) has been mentioned in line 176-184. They are repetitive.

We have removed it to avoid repetitive.

---

## Author Comment (AC2) · 3 Jun 2020

**Response to Referee #2**

We thank the reviewer very much for the detailed and valuable comments. We believe that addressing the issues raised by the reviewer will considerably improve the quality of our manuscript. Please see our response to each comment below (in blue).

This manuscript presents an overview of Global Estimates of Surface Reactive Nitrogen Concentration and Deposition Using Satellite Observation. The authors discuss recent advances of estimating surface Nr concentration and deposition, present a framework of using satellite data to estimate surface Nr concentration and deposition, and summarize the existing challenges for estimating surface Nr concentration and deposition using the satellite-based methods.

The manuscript is clearly written and logically organized. It provides sufficient and up-to-date literature citations. Listed below comments and suggestions for changes are relatively minor, but should be carefully considered. I recommend publication after addressing following comments:

1. L290: It is unclear to me how the vertical resolution of GEOS-Chem can resolve the vertical gradients that are likely to exist in source regions. The authors should clarify these several issues: (1) the vertical structure of the model, (2) the measurement characteristics of the surface observation (including height), (3) how this information is used to calculate surface concentrations.

IASI $NH_3$ retrievals are column data that has no vertical profile information. We gained surface $NH_3$ concentration by using modeled $NH_3$ vertical profiles from GEOS-Chem including 47 layers. We constructed the Gaussian model to fit the 47 layers' vertical $NH_3$ concentrations, which can generate the continuous $NH_3$ profile. Hence, based on the constructed the Gaussian model, we can obtain satellite-based

NH$_3$ concentration at any height. More importantly, the constructed the Gaussian model has general rules, appropriate for converting satellite columns to surface concentration simply. Please refer to the Sect. 3.1 for more details.

2. Fig. 10b: It is true that NH3 can be more accurately retrieved in one region than another depending on the thermal contrast. But it is not clear to me why this would be so much better in China than that in the US? I guess it is also just a matter of detection limits? It could also be related to more reliable simulation of mixing, depending on sufficient observational input into the parent weather model. Please clarify this issue.

We agree with you that the accuracy of IASI-retrieved surface NH$_3$ concentrations in different regions is highly linked with the thermal contrast (TC) and the simulation of NH$_3$ mixing from GEOS-Chem. The accuracy for satellite estimates over different area is related to the thermal contrast. The lowest uncertainties occurred when high columns and high TC coincide. In case either of them decreases, the uncertainty will gradually increase. In case both the TC and column are low, all sensitivity to NH$_3$ is lost. When high TC and high NH$_3$ columns (high HRI) occurs, the major contribution to the uncertainty results from the thickness of the NH$_3$ layer, the surface temperature as well as the temperature profile (Whitburn et al., 2016). We have added following text for clarification in the Sect. 4.2: "Higher correlation over China than other regions for the satellite estimates is linked to the detection limits by the instruments and thermal contrast (Liu et al., 2019).".

3. L531: For the estimated ammonia deposition, its uncertainties from remote sensing and models should be discussed more in this manuscript.

We have added the following text for further describing the uncertainties in the Sect. 4.2:

"The satellite NH$_3$ retrievals were affected by the detection limits of the satellite

instruments and thermal contrast. Higher accuracy could be gained with higher thermal contrast and $NH_3$ abundance. Instead, the uncertainties of $NH_3$ retrievals would be higher with lower thermal contrast and $NH_3$ abundance."

4. title: I suggest to change the satellite observation to "satellite retrievals" since IASI NH3 data were not a direct satellite observation but a reanalysis data using the statistical methods.

We have revised it as suggested.

5. L30: The abbreviation must be defined for the first occurrence.

We have removed these abbreviations.

6. L137: Replace ACTM with Atmospheric chemistry transport model

We have revised it as suggested.

7. L306: Added the references of the equations.

We have added the reference as suggested.

8. L333: Added the references of the equations.

We have added the reference as suggested.

---

## Author Response (AR2)

Dear Dr. Eliza Harris,

We appreciate the valuable suggestions given by you and the reviewers for improving the quality of our manuscript. In this document, we describe how we addressed your comments. Detailed responses to each comment are given below (in blue). We have addressed all the comments, and incorporated the comments or suggestions in the revised manuscript. Thank you very much for your consideration.

Sincerely,

Lei Liu

On behalf of all co-authors

This study reviews recent literatures on estimating reactive nitrogen (Nr) deposition using the satellite retrievals of NO2 and NH3, proposes a framework of using satellite data to estimate Nr deposition, and suggests a few research challenges. The topic of nitrogen deposition is important, and the compilation of recent literatures on reactive nitrogen deposition is useful to the research community. However, the manuscript mainly gives general descriptions of the previous results but lacks critical analysis and synthesis. The uncertainties in satellite measurements and chemical transport models, which are key to estimating Nr deposition based on satellite column measurements, are not addressed in detail. Overall, the scientific values of this work could be enhanced by more in-depth discussion of the advancement, challenges, and directions for future research.

The authors appreciate the valuable suggestions given by Referee #1 for improving the overall quality of the manuscript. In this document, we describe how we addressed the reviewer's comments. Detailed responses to each comment are given below (in blue).

**Specific comments:**

1. The authors highlight the advantages of satellite-based method compared to ground-based monitoring and ACTM simulation method. But there are significant uncertainties of satellite column measurements, especially for $NH_3$. In addition, the satellite-based method strongly depends on the ACTM simulation. What are the key uncertainties of the ACTM related to deposition estimates? How do the uncertainties in satellite measurements and ACTM affect satellite-based estimation? What are recommendations to reduce these uncertainties?

Yes, the uncertainties mainly came from the satellite retrievals and ACTM simulation. We did not aim to improve the accuracy of the satellite observations or the ACTM themselves, but to combine their advantages to gain surface $N_r$ concentrations with better performance with the ground-based measurements.

We have added the following text for more clarifications in the Sect. 4.2:

"For the dry deposition, the uncertainty mainly came from the satellite-derived estimates using the modeled vertical profiles. The uncertainty of vertical profiles modeled by CTM mainly resulted from the chemical and transport mechanisms. We recommend using the Gaussian function to determine the height of surface $NO_2$ and $NH_3$ concentrations that best matched with the ground-based measurements. There may exist systematic biases by simply using the relationship of $NO_2$ columns and surface concentration to estimate satellite surface $NO_2$ concentrations."

2. The authors propose a framework for combining satellite data, ground-based monitoring and ACTM (Figure 1). But it is not clear if it is a new idea. It seems that the approach has already been used in previous studies as indicated in the literatures shown in sections after Figure 1.

Yes, it's a new framework proposed in this study. Previous studies mainly focused on the methods to estimate surface $NO_2$ concentrations, while **Fig. 1** shows the general approach for estimating all $N_r$ spices on both concentration and deposition.

We have added the following text for further clarifications in the Sect. 3:

"Previous studies using satellite observation to estimate surface $N_r$ concentration and deposition only focused on one or several $N_r$ components, but not including all $N_r$ components, which were decentralized, unsystematic and incomplete. Here we give a framework of using satellite observation to estimate surface $N_r$ concentration and deposition as shown in **Fig. 1** based on recent advances.".

3. The title contains "Nr concentration and deposition", but the introduction part and the framework only mention "deposition". In my opinion, the estimation of Nr concentrations is just a part of the estimation of Nr depositions. There are many other studies which have offered more in-depth discussions of column concentrations of NO2 and NH3. I am not saying that concentrations cannot be shown but suggest framing the paper with a clearer focus on deposition.

Thanks for your suggestion. But, we think the introduction is appropriate since the estimation of $N_r$ concentrations is just a part of the estimation of dry $N_r$ depositions. The title included both the "$N_r$ concentration" and "deposition" because we reviewed on the methods of estimating both surface $N_r$ concentration and $N_r$ deposition.

We have added the following text for further clarifications in the introduction:

"Since the estimation of $N_r$ concentrations is just a part of the estimation of dry $N_r$ depositions, we here mainly reviewed the progress of dry $N_r$ depositions using the satellite observation.".

4. Line 193-195: Why may this method lead to an underestimation of surface NO2 concentration? In your proposed framework, the similar method has been used to estimate the surface NO2 concentration. Why is there no large underestimation in your validation? While you use the Gaussian function to fit the vertical concentration profile, but for the surface layer, you still use the ACTM derived the relationship between the NO2 column and surface NO2 concentration.

No, the methods in this study were different from the previous studies. We did not simply use the relationship between the $NO_2$ column and surface $NO_2$ concentration from the CTM. As presented in the main text, we can estimate surface $NO_2$ concentration at any height by using the Gaussian function. We used the surface $NO_2$ concentration at a certain height which best matched with the ground-based measurements.

We have added the following text for further clarifications in the Sect. 4.1:

"We did not simply use the relationship between the $NO_2$ column and surface $NO_2$ concentration from the CTM. As presented in the methods, we can estimate surface $NO_2$ concentration at any height by using the Gaussian function. We used the surface $NO_2$ concentration at a certain height (~60 m) which best matched with the ground-based measurements.".

5. Line 405-409: The derived NO2 columns from these satellites are quite different. Can you give some suggestions to the readers about which satellite data to use? Why do you choose OMI NO2 in your estimation? What are the results if you use other satellite data?

We have added the following text for further clarifications in the Sect. 4.1:

"The readers can use any satellite data combining the Gaussian function to estimate surface $NO_2$ concentrations. They can use surface $NO_2$ concentrations at a certain height which best matched with the ground-based measurements. The key is not selecting which satellite data we should use, but determining which height of surface $NO_2$ concentrations that better matched with the ground-based measurements by Gaussian function.".

6. Line 550-552: Can the similar method in equation 9 and 10 be used to estimate wet reduced Nr depositions? What are the different challenges for the estimations of wet reduced Nr depositions, compared with oxidized Nr?

Yes, the methods were the same for estimating both oxidized and reduced $N_r$ deposition. We did not identify big difference in the estimations of wet oxidized and reduced $N_r$ depositions.

We have added the following text for further clarifications in the Sect. 3.4:

"The mixed effects models were appropriate for estimating both wet $NO_3^-$ and $NH_4^+$ deposition using the satellite observations."

7. Section 5: For the trend estimation of Nr concentrations and depositions, have you conducted ACTM simulation for each year? The changes in emission and meteorology can significantly affect the Nr vertical profile and Nr species ratio, which are important in your satellite-based estimation.

Yes, we did. Please note that the simulated profile function has a general rule, which can be well simulated by Gaussian function for any year (for our case during 2005-2016). Thus, there is no need to simulate the vertical profile of $NO_2$ and $NH_3$ for each year.

We have added the following text for further clarifications in the Sect. 5:

"We used the proposed framework to estimate the long-term surface $NO_2$ concentrations by OMI during 2005-2016. Note that the simulated profile function has a general rule, which can be well simulated by Gaussian function for any year (for our case during 2005-2016).".

8. Line 567-569: This statement needs to be modified. As mentioned above, the satellite-based method strongly depends on the ACTM simulation. The uncertainties in emission inventories and other parts of ACTM can also significantly affect the vertical distribution of pollutants and the ratios of NO2 and other Nr species (e.g. HNO3, NH4+).

No, the emission inventories should not affect the vertical profiles shapes using Gaussian function, but the transport and chemical mechanism in the CTM may affect the accuracy of the vertical profile distribution. We mean that the satellite-based methods did not need to rely on the accuracy of the statistical emission data.

We have added the following text for further clarifications in the Sect. 5:

"The emission inventories should not affect the vertical profiles shapes using Gaussian function, but the transport and chemical mechanism in the CTM may affect the accuracy of the vertical profile distribution. The satellite-based methods did not need to rely on the accuracy of the statistical emission data."

9. Line 697: Are there any previous studies using a mechanism method to estimate Nr deposition?

We have added the following text for further clarifications in the Sect. 6:

"As far as we know, previous studies using satellite $NO_2$ and $NH_3$ column to estimate wet $N_r$ deposition were through a statistical way, and no studies were done from a mechanism perspective.".

**Minor comments:**

1. The authors should give the definition of reactive nitrogen (Nr). "Nr (such as NO3- and NH4+)" is mentioned in line 48, and "Nr (NOx and NH3)" is mentioned in line 59. This is confusing.

We have added the following text for clarifications:

"$N_r$ refers to the general term of N-containing substances in atmosphere, plants, soils and fertilizers that are not combined with carbon".

2. Line 57, change "mineral energy" to "fossil energy".

We have revised it as suggested.

3. Line 83, add "and" between the two words "accurate quantitative".

We have revised it as suggested.

4. Line 145-146: "Tian et al." should be "Tan et al. (2018)".

We have revised it as suggested.

5. Line 170: "Cheng et al. (Cheng et al., 2013)" should be "Cheng et al. (2013)". Please check the citation format throughout the manuscript.

We have checked the citation format throughout the manuscript as suggested.

6. Line 170-171: This sentence is not easy to understand. Please revise it.

We have revised it as follows:

"This method used the simple linear model and did not consider the vertical profiles of $NO_2$ (Cheng et al., 2013)"

7. Line 198-200: The study of Larkin et al., 2017 should be put in the previous paragraph discussing the method using the satellite data and statistical model. I think that the authors are discussing the method using the satellite data and ACTM-derived relationship in this paragraph.

No, Larkin et al. (2017) were also based on the satellite data and ACTM-derived relationship similar to Geddes et al. (2016), and it should be there.

8. Line 225-232: This information based on Jia et al. (2016) has been mentioned in line 176-184. They are repetitive.

We have removed it to avoid repetitive.

**Response to Referee #2**

We thank the reviewer very much for the detailed and valuable comments. We believe that addressing the issues raised by the reviewer will considerably improve the quality of our manuscript. Please see our response to each comment below (in blue).

This manuscript presents an overview of Global Estimates of Surface Reactive Nitrogen Concentration and Deposition Using Satellite Observation. The authors discuss recent advances of estimating surface Nr concentration and deposition, present a framework of using satellite data to estimate surface Nr concentration and deposition, and summarize the existing challenges for estimating surface Nr concentration and deposition using the satellite-based methods.

The manuscript is clearly written and logically organized. It provides sufficient and up-to-date literature citations. Listed below comments and suggestions for changes are relatively minor, but should be carefully considered. I recommend publication after addressing following comments:

1. L290: It is unclear to me how the vertical resolution of GEOS-Chem can resolve the vertical gradients that are likely to exist in source regions. The authors should clarify these several issues: (1) the vertical structure of the model, (2) the measurement characteristics of the surface observation (including height), (3) how this information is used to calculate surface concentrations.

We have added the following text for further clarification in the Sect. 3.1:

"Satellite $NO_2$ and $NH_3$ column data had no vertical profiles. Surface $NO_2$ and $NH_3$ concentration was estimated by modeled $NO_2$ and $NH_3$ vertical profiles from the CTM. The Gaussian model was constructed to fit the multiple layers' $NO_2$ and $NH_3$ concentrations with the altitude. The constructed Gaussian model has general rules, appropriate for converting satellite columns to surface concentration simply.".

2. Fig. 10b: It is true that NH3 can be more accurately retrieved in one region than another depending on the thermal contrast. But it is not clear to me why this would be so much better in China than that in the US? I guess it is also just a matter of detection limits? It could also be related to more reliable simulation of mixing, depending on sufficient observational input into the parent weather model. Please clarify this issue.

We agree with you that the accuracy of IASI-retrieved surface $NH_3$ concentrations in different regions is highly linked with the thermal contrast (TC) and the simulation of $NH_3$ mixing from GEOS-Chem. The accuracy for satellite estimates over different area is related to the thermal contrast. The lowest uncertainties occurred when high columns and high TC coincide. In case either of them decreases, the uncertainty will gradually increase. In case both the TC and column are low, all sensitivity to $NH_3$ is lost. When high TC and high $NH_3$ columns (high HRI) occurs, the major contribution to the uncertainty results from the thickness of the $NH_3$ layer, the surface temperature as well as the temperature profile (Whitburn et al., 2016).

We have added following text for clarification in the Sect. 4.2:

"Higher correlation over China than other regions for the satellite estimates is linked to the detection limits by the instruments and thermal contrast (Liu et al., 2019).".

3. L531: For the estimated ammonia deposition, its uncertainties from remote sensing and models should be discussed more in this manuscript.

We have added the following text for further describing the uncertainties in the Sect. 4.2:

"The satellite $NH_3$ retrievals were affected by the detection limits of the satellite instruments and thermal contrast. Higher accuracy could be gained with higher thermal contrast and $NH_3$ abundance. Instead, the uncertainties of $NH_3$ retrievals would be higher with lower thermal contrast and NH$_3$ abundance."

4. title: I suggest to change the satellite observation to "satellite retrievals" since IASI NH3 data were not a direct satellite observation but a reanalysis data using the statistical methods.

We have revised it as suggested.

5. L30: The abbreviation must be defined for the first occurrence.

We have removed these abbreviations.

6. L137: Replace ACTM with Atmospheric chemistry transport model

We have revised it as suggested.

7. L306: Added the references of the equations.

We have added the reference as suggested.

8. L333: Added the references of the equations.

We have added the reference as suggested.

[revised manuscript text omitted]